# Changes in the Aroma Profile and Phenolic Compound Contents of Different Strawberry Cultivars during Ripening

**DOI:** 10.3390/plants13101419

**Published:** 2024-05-20

**Authors:** Kristyna Simkova, Robert Veberic, Mariana Cecilia Grohar, Massimiliano Pelacci, Tina Smrke, Tea Ivancic, Aljaz Medic, Nika Cvelbar Weber, Jerneja Jakopic

**Affiliations:** 1Department of Agronomy, Biotechnical Faculty, University of Ljubljana, Jamnikarjeva 101, 1000 Ljubljana, Slovenia; robert.veberic@bf.uni-lj.si (R.V.); jerneja.jakopic@bf.uni-lj.si (J.J.); 2Agricultural Institute of Slovenia, Hacquetova Ulica 17, 1000 Ljubljana, Slovenia

**Keywords:** *Fragaria* × *ananassa*, volatile organic compounds, phenolics, fruit development, cultivar

## Abstract

Secondary metabolites, namely, phenolic and volatile organic compounds, contribute to the nutritional and organoleptic quality of the strawberry fruit. This study focuses on the changes in the content of phenolic compounds and volatile organic compounds during the ripening, from green to overripe fruit, of five strawberry cultivars (‘Asia’, ‘CIVN 766’, ‘Aprica’, ‘Clery’, and ‘Malwina’). Additionally, these changes are compared with the colour of the fruit and peroxidase and polyphenol oxidase activity. Our results show that the accumulation of secondary metabolites (phenolic and volatile organic compounds) significantly changed during the ripening process for all of the studied cultivars. As for phenolic compounds, flavanols and hydroxybenzoic acid derivatives comprised between 87 and 95% of the total phenolic compound content in unripe green fruit. In contrast, anthocyanins and hydroxycinnamic acid derivatives comprised between 64 and 77% of the total phenolic compound content in overripe fruit, except in the fruit of the cultivar ‘CIVN766’. When it comes to the aroma profile, the content of aldehydes decreased by 24–49% as the fruit ripened, and the accumulation of esters increased. Our study also shows that the ripening process differs among cultivars, and it is therefore necessary to define ripening indicators separately for each cultivar.

## 1. Introduction

Strawberries contain a wide range of secondary metabolites, contributing to their nutritional and organoleptic quality. The major non-volatile secondary metabolites found in strawberries include phenylpropanoids (mainly derivatives of cinnamic acid, 4-coumaric acid, caffeic acid, and ferulic acid), flavonoids (mainly anthocyanins, glycosides of quercetin and kaempferol, derivatives of catechin and epicatechin), and ellagic acid derivatives [1]. Out of these compounds, anthocyanins (mainly derivatives of pelargonidin and cyanidin) are the main pigments in strawberries, and their content can affect the colour of the fruit, which is an important quality parameter [2]. The characteristic red colour develops during ripening through the production of anthocyanins, which reaches a peak at full ripening [2,3,4]. Furthermore, phenolic compounds contribute to the nutritional value of the fruit. According to epidemiological studies, it is believed that polyphenolic compounds in fruit contribute to lowering the risk of chronic diseases such as cancer and cardiovascular diseases, as well as to lowering the risk of Alzheimer’s disease [5,6,7,8]. These health benefits can even further enhance consumer demand.

In addition to non-volatile secondary metabolites, plants also produce volatile metabolites [1]. These volatile organic compounds are significant components of fruit flavour, and even slight changes modify the taste, even though these compounds account for 0.001–0.01% of the fruit’s weight [9]. According to a review by Ulrich et al. [10], 979 volatiles have been identified in the strawberry fruit. Different groups of volatile compounds can be found in strawberries, including esters, furans, terpenoids, aldehydes, alcohols, ketones, acids, lactones, sulphur compounds, and acetals [11]. Esters are a key group of volatile organic compounds present in ripe strawberries (comprising up to 90% of the total number of volatile organic compounds) [12]. While esters are important aroma compounds in ripe strawberries, aldehydes and alcohols are important volatiles responsible for green, unripe notes [13]. However, the profile of these volatile organic compounds, as well as of phenolic compounds, changes during ripening [13,14,15,16,17,18,19,20,21], but the content of secondary metabolites is specific to each cultivar [12,22,23]. Most previous studies focus on one cultivar, a limited number of ripening stages, or one group of secondary metabolites. Therefore, there is a need to study the overall changes in the content of secondary metabolites in more cultivars grown under the same conditions in order to understand these changes and identify markers which can help in monitoring the ripening process and determining an optimal harvest point.

This study aimed to describe the changes in the aroma profile and phenolic compound contents during the ripening process of five different strawberry cultivars and determine if there is any relation between these changes and the colour parameters. This study included five June-bearing cultivars (‘Asia’, ‘Clery’, ‘Aprica’, ‘CIVN 766’, and ‘Malwina’) which were sorted into five ripening stages: green fruit, white fruit, ripe red, fully ripe dark red, and overripe fruit. Additionally, the enzyme activity of peroxidase and polyphenol oxidase was measured to see if it has any impact on the content of phenolic compounds. The results could further serve as a basis for defining fruit ripening indicators that can be used by producers and also by the processing industry.

## 2. Results

### 2.1. Colour

The lightness (L*), hue angle (h°), and b* parameter of the fruit decreased during the ripening process, and the a* parameter increased (Table 1) when comparing green fruit (R1) and ripe fruit (R3–R5). The L* value ranged between 53.7 and 59.8 in unripe green fruit (R1) among the cultivars, while in fully ripe fruit (R4), this value ranged between 22.3 and 30.2. The a* parameter ranged between 1.3 and 2.0 in unripe green fruit (R1), while in fully ripe fruit (R4), it ranged between 20.7 and 36.9. As for the b* parameter, this value ranged between 30.9 and 35.5 in unripe green fruit (R1) and between 11.9 and 23.6 in the fully ripe fruit (R4). However, in most cultivars, except for ‘Clery’, the a* value decreased from early ripe fruit (R3) to overripe fruit (R5). The overripe fruit (R5) of the cultivar ‘Malwina’ reached the darkest (lowest L* parameter) and the least intense colour (lowest C*) out of the studied cultivars. This can also be observed in the fruit pictures presented in Appendix A. Among the cultivars, the highest intensity of colour (highest C*) was primarily observed in the middle of the ripening at the ripe stage (R3), except for the fruit of the cultivars ‘Malwina’ and ‘Aprica’. The fruit of the cultivar ‘Aprica’ had comparable levels of colour intensity in both the ripe and fully ripe fruit (R3 and R4), and in the fruit of the cultivar ‘Malwina’, the first three ripening stages (R1–R3) had comparable colour intensity. The cultivar had a significant effect on all of the measured colour parameters, but the variance based on the ripening stage was higher.

### 2.2. Phenolic Compounds

The profile of the phenolic compounds differed among the ripening stages, and the complete list of identified phenolic compounds for each cultivar can be found in Appendix A. As shown in Table 2, the total phenolic compound content also differed among the ripening stages. In most cultivars, the phenolic compound content was higher at the overripe (R5) stage than at the green (R1) stage, with the exception of the cultivar ‘Aprica’, where the content was higher at the beginning of the ripening (R1) stage than in the other ripening stages. Additionally, in the fruit of the cultivars ‘Asia’, ‘CIVN766’, and ‘Malwina’, the total phenolic compound content decreased after the initial ripening stage, and the lowest content was detected at the white stage (R2) and, in the case of ‘CIVN766’, also at the ripe stage (R3). In the fruit of the cultivar ‘Clery’, the total phenolic compound content was comparable between the first two ripening stages (R1 and R2), and it gradually increased with the following ripening stages.

These differences could be attributed to differences in the contents of the specific groups of phenolic compounds. While in the initial stage, the major phenolic compounds were flavanols and hydroxybenzoic acid derivatives, the content of anthocyanins and hydroxycinnamic acid derivatives increased during the ripening process. In most of the cultivars in the three last ripening stages (R3–R5), anthocyanins comprised the majority of the phenolic compound content, except for the fruit of the cultivar ‘CIVN766’, where the hydroxybenzoic derivatives were the major group of phenolic compounds present in the fruit. The anthocyanin content significantly differed among the last three ripening stages (R3–R5). The anthocyanin content was the highest in the overripe ripening stage (R5) for the cultivars ‘Malwina’ and ‘Clery’. For the fruit of the cultivars ‘Asia’ and ‘CIVN766’, the anthocyanin content differed only between the early ripe fruit (R3) and the fully ripe (R4) or overripe (R5) fruit, but no significant difference was found between the last two ripening stages (R4 and R5). In the fruit of the cultivar ‘Aprica’, the anthocyanin content in the overripe fruit did not significantly differ from the other two ripe stages (R3 and R4). In addition to the effect of the ripening stage on the contents of phenolic compounds, the cultivar also had a significant effect on the contents of all phenolic groups. The content of anthocyanins, flavanols, and hydroxycinnamic acid derivatives correlated with some of the colour parameters (Appendix A). As the L* parameters decreased and the fruit became darker, the content of anthocyanins and hydroxycinnamic acid derivatives increased, with correlation coefficients r equal to −0.82 and −0.73, respectively. As the a* parameter increased and the fruit became redder, the content of flavanols decreased (r = −0.67). Additionally, as the fruit colour intensity (C*) increased, the content of flavanols decreased, with a correlation coefficient of −0.73.

#### Anthocyanin Content

Among the five studied cultivars, 10 different individual anthocyanins were identified in the fruit (Appendix A). In all cultivars, pelargonidin-3-*O*-glucoside had the highest content in the ripe fruit (R3–R5), as presented in Table 3. In the first two ripening stages, no or a very low anthocyanin content was detected, and this could be attributed to the content of cyanidin-3-*O*-galactoside, pelargonidin-3-*O*-glucoside, or cyanidin-3-(6″malonyl)glucoside. Among the ripe fruit (R3–R5), the content of the individual anthocyanins was mostly higher in the last two ripening stages (R4 and R5) compared to the early ripe fruit (R3) in the fruit of the cultivars ‘Malwina’ and ‘Asia’. A significant increase between the overripe stage and the other ripe stages (R3 and R4) was detected in the content of most of the individual anthocyanins in the fruit of the cultivar ‘Malwina’, except for cyanidin-3-*O*-galactoside, cyanidin-3-(6″malonyl)glucoside, and pelargonidin-3-*O*-acetylglucoside, where the content was comparable between overripe (R5) and fully ripe fruit (R4). However, not all cultivars showed similar trends. For example, the fully ripe (R4) and overripe fruit (R5) of the cultivar ‘CIVN766’ showed comparable contents of all detected individual anthocyanins. The cultivar displayed significant effects on the content of all individual anthocyanins, except for the pelargonidin derivative. The content of some of the individual anthocyanins also correlated with the colour parameters (Appendix A). Out of the anthocyanins, cyanidin-3-*O*-galactoside, cyanidin-3-*O*-glucoside, pelargonidin-3-*O*-glucoside, pelargonidin-3-*O*-rutinoside, and cyanidin-3-(6″malonyl)glucoside showed a significant correlation with the majority of the measured colour parameters. As the fruit became darker, the lightness parameter (L) and the b* value decreased, and the content of all of the above-mentioned individual anthocyanins increased. Additionally, as the content of 5-pyranopelargonidin-3-glucoside increased, the a* value decreased (r = 1.00). The content of cyanidin-3-*O*-galactoside and cyanidin-3-(6″malonyl)glucoside showed a positive correlation with the a* parameter, with correlation coefficients of 0.31 and 0.54, respectively, consequently contributing to the red colour of the fruit. The content of cyanidin-3-*O*-galactoside, cyanidin-3-*O*-glucoside, and pelargonidin-3-*O*-rutinoside negatively correlated with the colour intensity (C*), with correlation coefficients of −0.45, −0.72, and −0.96, respectively.

### 2.3. Enzyme Activity

The enzyme activity of peroxidase (POD) and polyphenol oxidase (PPO) also showed significant differences among the ripening stages (Table 4). The POD activity increased in all studied cultivars during ripening. Significant differences in the POD activity were observed between the initial ripening stage (R1) and the overripe stage (R5), and there were significant differences found even between the early ripe fruit (R3) and the overripe fruit (R5) of most cultivars, except for the fruit of the cultivar ‘Asia’. On the other hand, the PPO activity decreased during ripening, and the last three ripening stages (R3–R5) showed significant differences only in the fruit of the cultivars ‘Aprica’ and ‘Asia’. For both enzymes, significant variance was observed not only based on the ripening stage but also based on the cultivar. The enzyme activity was not shown to be negatively affected by the content of phenolic compounds, as there was no negative correlation found between the enzyme activity and the content of the phenolic compounds (Appendix A). However, a positive correlation was found between PPO activity and flavanol content (0.75). Additionally, a positive correlation was found between POD activity and the content of hydroxycinnamic acid derivates (0.49), as well as with the content of anthocyanins (0.57) and the total phenolic compound content (0.48).

### 2.4. Aroma Profile

While at the beginning of the ripening process, aldehydes comprised the majority of the volatile organic compounds (VOCs) and esters were not detected or detected in very small amounts, at the end of the ripening process, the content of esters comprised the majority of the present VOCs (Table 5). Additionally, the content of terpenoids also increased during the ripening, and in the fruit of the cultivar ‘CIVN766’, mesifurane was detected in the last two ripening stages (R4 and R5). The content of alcohols remained stable or decreased during ripening. The exception was the cultivar ‘Clery’, where the content of alcohols significantly increased at the end of the ripening, which can be attributed to the presence of 2-hexen-1-ol at the overripe stage (R5) (Appendix A). Significant differences were also observed among the last three ripening stages (R3–R5). For most of the cultivars, except for ‘Clery’, the content of esters continued to increase. The changes in the VOCs present at each ripening stage and the contents of the individual compounds can be found in Appendix A.

A PLS-DA (partial least-square discriminant analysis) was performed to obtain an overview of the differences between the different ripening stages and to determine whether creating a model to discriminate between different ripening stages based on their aroma profile would be possible. Using overfitting, we were able to create a model that can explain 73.9% of the data, but 12 components had to be used to achieve it (Figure 1A). Using only two components, the model would be able to explain only 32.9% of the data. This could be due to the similarity between some of the ripening stages, since when the data for ripening stages R2 and R4 were excluded from the analysis, a model that explains 71.6% of data could be created using only two components. The PLS-DA model performs better when limited to one cultivar. In our case, when limited to one cultivar, the PLS-DA model can explain between 77.8 and 86.6% of the data using just four components.

We can see from the loading plot (Figure 1B) that the content of certain aldehydes (marked with AD), such as pentanal (AD1), heptanal (AD5), hex-(2Z)-enal (AD6), hex-(2E)-enal (AD7), and pent-(2E)-enal (AD3), together with 1-penten-3-ol (AC1), contributes to distinguishing unripe fruit. On the other hand, the content of certain esters (marked as E), such as isopropyl butyrate (E3), 3-methylbutyl butanoate (E17), 1-methylhexyl butanoate (E20), and ethyl isovalerate (E25), contributes to discriminating overripe fruit. A complete list of codes for individual aroma compounds can be found in Appendix A. Using the permutation test (Figure 1C), we validated the model, and as seen on the ROC plot (Figure 1D), the model can effectively distinguish between the different ripening stages.

### 2.5. Principal Component Analysis (PCA)

A PCA of all of the measured parameters (Figure 2) showed that the samples could be clustered into different groups. The green and white fruit of all cultivars differed from the other samples due to their higher contents of flavanols, flavonols, hydroxybenzoic acid derivatives, alcohols, and aldehydes, their higher PPO activity, and their higher values of L*, b*, and h°. The fully ripe and overripe fruits of the cultivar ‘Malwina’ stand apart from the samples due to their higher total content of phenolic compounds, higher content of anthocyanins, and higher total content of terpenoids. The ripe fruits (R3–R5) of the cultivars ‘Aprica’ and ‘Clery’ stand out due to their higher total content of VOCs, higher content of esters and higher values of C* and a*.

## 3. Discussion

The sensorial and nutritional quality of strawberries is an important factor in current strawberry breeding strategies [24], and the content of both phenolic and volatile organic compounds contributes to this. However, as shown in this study, the content of these compounds can vary based on the cultivar and the ripening stage.

In most cultivars, the total content of phenolic compounds increased during ripening, which was previously observed in other studies [15,25]. However, the fruit of the cultivar ‘Aprica’ showed the highest content in the initial green ripening stage, and ‘Malwina’ and ‘CIVN766’ showed a high phenolic compound content in the green stage, comparable to the phenolic compound content in the ripe fruit. This could be attributed to the high content of hydroxybenzoic acid derivatives, mainly ellagitannins, coming from the achenes [26], which make up a significant proportion of the fruit at the early stage of ripening. Our study also showed that this can vary depending on the cultivar, which can be related to the achene per flesh ratio of the cultivars. Additionally, it has previously been reported that the ellagic acid content is high in unripe strawberry fruit and decreases during ripening [4,16]. Additionally, as shown in our study as well as other studies [26], unripe fruit also contains a high amount of proanthocyanidins (flavanols). The higher content of flavanols and ellagic acid derivatives could thus serve in unripe fruit as a protection mechanism against feeding and pathogenic attacks [27]. On the other hand, as the fruit ripened, the content of hydroxycinnamic acid increased, similarly to previous studies [4,17,26], where the content of *p*-coumaric acid and cinnamic acid derivatives increased during ripening. This can be explained by the higher expression of the genes involved in the metabolic pathways leading to the synthesis of the glucose esters of these phenolic acids, as previously reported by Lunkenbein et al. [28]. Similarly, as previously reported [3,17], the content of flavonols did not follow a specific trend during ripening, which suggests that other factors affect the biosynthesis of these compounds, such as environmental effects [3].

The content of anthocyanins as one of the major phenolic compound groups and the main pigments in strawberries also increased during ripening, which is in agreement with previous studies [4,17,26]. During fruit ripening, the content of chlorophyll decreases [29] and the content of anthocyanins increases, as also confirmed by our results [29]. The higher accumulation of anthocyanins can be attributed to increased gene expression in the anthocyanin biosynthetic pathways [30]. These changes are also reflected in the colour of the fruit. While at the beginning of the ripening, the a* value was low (almost towards a green colour), in the ripe stages of the fruit (R3–R4), the a* parameter reached values up to 38.6. Most of the individual anthocyanins were detected mainly in the last three ripening stages and, in many cases, showed an increase in the content from early ripe fruit to overripe fruit, which shows that all of the individual anthocyanins are regulated in the same way. As pigments, the content of anthocyanins also affects the colour of the fruit. The content of cyanidin-3-(6″malonyl)glucoside contributed positively to the redness of the fruit (a* parameter). On the other hand, the content of cyanidin-3-*O*-glucoside, which contributes to the dark red colour of the fruit [2], correlated negatively with redness (a*) and also contributed to the darker tones (lower L* value). Additionally, pelargonidin-3-*O*-rutinoside was shown to contribute negatively to the red colour notes of the fruit, similarly to a previous study [31]. Out of all the identified anthocyanins, cyanidin-3-*O*-galactoside, cyanidin-3-*O*-glucoside, pelargonidin-3-*O*-glucoside, pelargonidin-3-*O*-rutinoside, and cyanidin-3-(6″malonyl)glucoside were found to affect the colour of the fruit significantly.

The enzyme activity of peroxidase and polyphenol oxidase is an important fruit quality parameter, as these enzymes are involved in enzymatic browning [32]. Additionally, peroxidase plays an important role in regulating the ripening process [33]. As previously reported [34,35], the polyphenol oxidase (PPO) activity decreases at the initial stages of ripening, which was also observed in our study in most of the studied cultivars. The exception is the cultivar ‘Malwina’, but as mentioned by Jia at al. [35], the PPO activity can also increase at later stages of ripening, and the activity depends on the pH. This shows that the changes in PPO activity depend not only on the ripening stage but also on the cultivar and the pH of the fruit. While the PPO activity decreased, the peroxidase (POD) activity increased. This is in contrast with previous studies [33,36], where the POD activity was higher in the initial stages. This could be attributed to a difference in the POD activity among the cultivars and the different content of metabolites present in the fruit at each ripening stage, which can affect the enzyme activity. Although the POD and PPO activity did not show a negative effect on the content of phenolic compounds, a positive correlation between the enzyme activity and the content of anthocyanins, flavanols, hydroxycinnamic acid derivatives, and the total phenolic content was observed. Since polyphenol oxidase is located in chloroplast thylakoid membranes, and its phenolic substrates are located in the vacuoles [32], the enzyme cannot access its substrates while the fruit is still whole. Additionally, peroxidase activity is limited by the availability of electron acceptor compounds such as hydrogen peroxide [32]. Therefore, as confirmed by this study, these enzymes do not negatively affect the phenolic compound content during ripening.

Another important group of secondary metabolites is volatile organic compounds (VOCs), which comprise the aroma profile of the fruit and contribute to its organoleptic quality [37]. As previously observed [12,13,20,38], the content of aldehydes decreased during ripening, and the content of esters increased. While aldehydes remained present during the whole process of ripening and could contribute to the green aroma notes in the ripe fruit, esters were not detected at the early stages of ripening of some of the studied cultivars. The accumulation of aldehydes and alcohols in the initial stages of ripening can be attributed to the breakdown of long fatty acids by lipoxygenase (LOX) [39], which produces aldehydes and alcohols that can then serve further ester synthesis. The increase in the content of esters could be attributed to the lack of esterase activity in the unripe fruit [12]. The enzyme responsible for the final step of ester formation is alcohol acyltransferase (AAT), and a correlation has been previously observed between the activity of this enzyme and the content of some esters [39]. Additionally, branched amino acids can also serve as substrates for the synthesis of branched-chain volatiles in strawberry fruit [39], which, in our case, occurred mainly towards the end of the ripening period. As for the individual compounds, based on the PLS-DA, 1-penten-3-ol, pentanal, heptanal, hex-(2Z)-enal, hex-(2E)-enal, and pent-(2E)-enal can all serve as indicators of unripe fruit. This is similar to a previous study including nine strawberry cultivars [13], where 1-penten-3-ol and 2-hexenal were among the compounds identified as indicators of unripe fruit. On the other hand, the majority of the detected esters and terpenes can serve as indicators of ripe fruit. Our results are in line with a previous study on the cultivar ‘Candoga’, where an increase in the content of methyl butyrate and methyl hexanoate was observed [18]. Furthermore, the content of esters such as ethyl hexanoate, hexyl acetate, and terpenes can also have a positive effect on consumer acceptance, and thus, it is necessary to analyse and monitor the contents of these esters during the ripening process [18]. As for overripe fruit, isopropyl butyrate, 3-methylbutyl butanoate, 1-methylhexyl butanoate, or ethyl isovalerate can serve as possible indicators. Additionally, in the cultivar ‘CIVN766’, mesifurane is a good indicator for ripe and overripe fruit. Mesifurane synthesis depends on the presence and activity of a specific enzyme, *O*-methyltransferase [40], and based on our results, this enzyme is active mainly in this cultivar. Among these, mesifurane and isopropyl butyrate were previously identified as compounds that negatively affect consumer acceptance [18]. This means that the accumulation of some of the compounds towards the end of the ripening can negatively affect their organoleptic quality. Moreover, we can see from the PLS-DA that there is potential for creating a reliable model for determining the ripeness of the fruit based on its aroma profile. However, there are still some challenges, as it is difficult to build a model that covers a range of cultivars with different aroma profiles, and it can be difficult to distinguish between ripening stages which are similar to each other, as shown in our results.

The PCA shows that while the unripe fruits of different cultivars share similar attributes, the ripe and overripe fruits of different cultivars may have different indicators. This can be observed mainly for the cultivar ‘Malwina’, which clusters separately from the ripe and overripe fruits of other cultivars. Based on the obtained results, it is evident that drawing generalisable conclusions regarding strawberries is not feasible. Due to the observed complexity of the secondary metabolism during ripening, a meticulous examination of each individual variety is required. All of these changes should also be combined with the change in other physical characteristics (size and firmness) and the content of primary metabolites (sugars and acids). It has previously been reported [29] that the fruit becomes softer during ripening, the content of acids decreases, and the content of sugars increases. All of these changes contribute to the overall quality of the fruit and, consequently, to consumer acceptance.

## 4. Materials and Methods

### 4.1. Plant Material

The following five economically important strawberry cultivars were obtained for this experiment: ‘Asia’, ‘CIVN 766’, ‘Aprica’, ‘Clery’, and ‘Malwina’. Each cultivar has its own specific fruit colour: light red, almost orange in the case of ‘CIVN 766’, light red in the cultivar ‘Clery’, red in ‘Asia’ and ‘Aprica’, and dark red in the case of the ‘Malwina’ cultivar. The cultivars ‘Clery’, ‘Asia’, and ‘Aprica’ originated in Italy and were registered in 2002, 2005, and 2015, respectively. The cultivar ‘Malwina’ originated in Germany (Kraege International, Telgte, Germany), and crossbreeding was performed in 1998. The cultivar ‘CIVN 766’ originated in Italy (Consorzio Italiano Vivaisti, Comacchio, Italy) and is currently being tested. Fruit samples were collected during the peak ripening time for each cultivar, which occurred throughout June 2021. The fruit samples were harvested at the orchard situated at the Research Station of the Agricultural Institute of Slovenia in Brdo pri Lukovici, which is located at latitude 46°10′ N and longitude 14°41′ E. Each cultivar was planted in five blocks, each containing 10 plants (totalling 50 plants per cultivar). The plants were placed on slightly elevated beds covered with black polyethylene. They were arranged in double rows with a spacing of 0.25 × 0.25 m between plants within the bed and 1.3 m between adjacent beds. The average temperature inside the tunnel fluctuated between 8.5 °C and 22.1 °C during the production period, with light levels ranging from 0 to 643.2 W m^−2^ and humidity levels varying between 63.2% and 96.4%. The harvest dates and growing conditions were the same as previously described by Simkova et al. [29]. Each strawberry cultivar was harvested in five different fruit maturity stages during the ripening time: green fruit (R1), white fruit (R2), ripe red fruit (R3), fully ripe dark red fruit (R4), and overripe fruit (R5), as shown for the example of the ‘Aprica’ cultivar (Figure 3). Pictures of the other cultivars can be found in Appendix A. The samples were immediately transported to the laboratory for colour measurement and sample preparation.

### 4.2. Colour

From each ripening stage of every cultivar, 14 fruits were picked, and their colour was measured. The colour parameters were measured in the CIELAB colour space using a colourimeter, CR-10 Chroma (Minolta, Osaka, Japan). The colour parameters included the L* value, indicating the level of lightness (0 represents black and 100 represents white); the h° value, indicating the colour in degrees (0° corresponds to red, 90° corresponds to yellow, 180° corresponds to green, and 270° corresponds to blue); the C* value, indicating the chroma (higher values indicate a more intense colour); the a* value, representing the colour along the green–red axis; and the b* value, representing the colour along the yellow–blue axis.

### 4.3. Dry Matter

The dry matter content was determined for each ripening stage and cultivar separately. The samples were dried in the oven at 105 °C for 72 h. The measurement was performed in 6 replicates for each sample type, and the results were used to calculate the phenolic compound content per dry weight.

### 4.4. Sample Preparation

For each extraction, 6 repetitions were prepared and pooled from 14 berries in order to ensure representative results. Samples for phenolic compound extraction were frozen in liquid nitrogen and stored at −20 °C until further analysis. Samples for enzyme activity measurements and aroma profile analysis were frozen in liquid nitrogen and stored at −80 °C.

### 4.5. Analysis of Anthocyanins and Other Phenolic Compounds

The extraction of phenolic compounds followed the procedure described by Simkova et al. [41]. The identification of phenolic compounds was performed using available standards and the fragmentation pattern obtained from an LTQ XL mass spectrometer (Thermo Scientific, Waltham, MA, USA) following the procedure described by Simkova et al. [41]. The phenolic compound content was determined using a Dionex UltiMate 3000 HPLC (Thermo Scientific, USA) system with the Gemini C18 column (150 × 4.6 mm, 3 µm; Phenomenex, Torrance, CA, USA). The sample response was measured at 280, 350, and 530 nm using the same conditions described by Simkova et al. [41]. The flow rate was 0.6 mL min^−1^, and the total duration of the analysis was 50 min. The mobile phase gradient was as follows: 5% solvent B from 0 to 15 min, 5–20% B from 15 to 20 min, 20–30% B from 20 to 30 min, 30–90% B from 30 to 35 min, 90–100% B from 35 to 45 min, and then, 100–5% solvent B from 45 to 50 min. Solvent A was 3% acetonitrile and 0.1% formic acid in bidistilled water (*v*/*v*/*v*), and solvent B was 3% bidistilled water and 0.1% formic acid in acetonitrile (*v*/*v*/*v*).

The phenolic compound content was calculated according to a corresponding external standard or structurally similar compound. The content was expressed as the mg equivalent of this compound per kg dry weight. The external standards used for quantification were as follows: cyanidin-3-*O*-glucoside, procyanidin B1, ellagic acid, p-coumaric acid, ferulic acid, apigenin-7-glucoside, ellagic acid, caffeic acid, kaempferol-3-glucoside, and quercetin-3 glucoside from Fluka Chemie (Buchs, Switzerland); pelargonidin-3-*O*-glucoside from Sigma-Aldrich (Steinheim, Germany); and isorhamnetin-3-glucoside from Extrasynthese (Genay, France).

### 4.6. Enzyme Activity

The enzyme extraction and assays followed the procedure described by Simkova et al. [41] with optimisations for microplate measurements.

The fruit was ground to a powder using an analytical mill (IKA A11 basic, Staufen, Germany) with liquid nitrogen. The sample powder (1 g) was mixed with 0.5 g of Polyclar, 0.5 g of sand, and 4 mL of extraction buffer (0.01 M TRIS, 0.007 M EDTA and 0.01 M Borax). This mixture was vortexed for 30 s and centrifuged for 10 min at 10,000 rpm at 4 °C (Eppendorf Centrifuge 5810 R, Hamburg, Germany). Before measurement, the extract (400) µL was cleaned through Sephadex G-25 gel columns. The assays were measured on a Synergy HTX multi-mode reader (Agilent Technologies, Santa Clara, CA, USA), and data were collected using Gen5 software (version 3.12.08). The POD assay consisted of 180 µL of H_2_O_2_-KPi buffer, 20 µL of the sample extract, and 20 µL of a 0.004 M *o*-dianisidine solution in methanol and was measured for 20 min at 460 nm. The PPO assay consisted of 100 µL of McIlvaine buffer (pH 6.5), 75 µL of the sample extract, and 70 µL of a 0.2 M pyrocatechol solution and was measured for 15 min at 410 nm. The maximum absorbance change per minute was calculated as an average from the three maximum values.

The protein content was measured using the Bradford method following the protocol defined by Kruger [42]. The final enzyme activity is presented as the change in absorbance per minute and protein content in the enzyme assay (A min^−1^ mg^−1^ protein).

### 4.7. Aroma Profile Analysis

The volatile organic compound profile was analysed through gas chromatography analysis (HS-GC-MS) based on the method described by Baluszynska et al. [43] with the modifications described in this section. Strawberry samples (stored at −80 °C) were ground to a fine powder using an analytical mill (IKA A11 basic, Staufen, Germany) with liquid nitrogen. The powder (2 g) was placed in 20 mL vials, and 10 µL of an internal standard (methyl nonanoate, 1:100, 11 mg mL^−1^ in acetonitrile) was added. The mixture was added to a closed vial with a screw cap with a PTFE-silicon septum and transferred to a Shimadzu AOC-20s autosampler, where it was incubated at 65 °C for 20 min with constant shaking at 250 rpm. A volume of 1000 μL of the headspace portion was injected in a splitless mode for 0.4 min into the injection port at 250 °C at a 25 mL/min injection rate. A Shimadzu GC-MS QP2020 gas chromatography system with a Single Quadropole MS detector was used. A ZB-wax PLUS capillary column (30 m × 0.25 mm, 0.5 μm film thickness) was used to separate the volatile organic compounds. The carrier gas was helium with a 1 mL/min flow rate. The temperature program was set as follows: first, the temperature was held at 45 °C for 3 min, then raised to 110 °C at a rate of 10 °C/min, then raised to 150 °C at 20 °C/min and held for 5 min, then raised to 240 °C at 15 °C/min and, finally, held at 240 °C for 2 min. The temperature of the interface and the MS ion source was set at 240 °C, the scan rate was set at 2.0 scan/s, the ionisation energy was set at 70 eV, and the mass scan range was set at 50–500 *m*/*z*. Volatile organic compounds of the samples were identified based on their retention times (Rts) and commercial libraries of spectra (NIST 11 and FFNSC 4), and they were semi-quantified based on each compound and internal standard peak areas and the internal standard and sample weight. Sensory descriptors were obtained from the Flavornet and FlavorDB databases [44,45].

### 4.8. Statistical Analysis

The data were analysed statistically in R x64 4.1.2 using the Rcmdr package. The data are expressed as means ± standard errors, and the results are presented as the mean of the two harvest points. Significant differences among the ripening stages within each cultivar were determined by means of one-way analysis of variance (ANOVA) with Duncan’s test. Multiway ANOVA was performed to evaluate the effect of the cultivar, ripening stage and their interaction on the results. Pearson correlation was used to assess the relationships between the different parameters. A significant difference, effect or correlation was considered at *p* < 0.05. The partial least-squares discriminant analysis (PLS-DA) was performed using SIMCA 18 software (Sartorius, Umeå, Sweden) using auto-fitting. A permutation test (*n* = 100) was completed to further validate the model. The principal component analysis (PCA) was performed in R using the FactoMineR package [46]. For PLS-DA and PCA, any compounds detected at a certain stage were replaced with 0.

## 5. Conclusions

Our study found significant changes in the secondary metabolite content during strawberry ripening. While the content of flavanols and hydroxybenzoic acid derivatives decreased during ripening, the content of hydroxycinnamic acid derivatives and anthocyanins increased. Moreover, individual anthocyanins such as cyanidin-3-*O*-galactoside, cyanidin-3-*O*-glucoside, pelargonidin-3-*O*-glucoside, pelargonidin-3-*O*-rutinoside, and cyanidin-3-(6″malonyl)glucoside notably influenced the colour of the fruit. Peroxidase activity increased during ripening, while polyphenol oxidase activity decreased, except for ‘Malwina’. The enzyme activity did not negatively affect the content of phenolic compounds. In addition to the phenolic compound profile, the aroma profile and the content of volatile organic compounds changed during ripening. While the content of aldehydes decreased, the content of esters and terpenoids increased. Compounds such as 1-penten-3-ol, pentanal, heptanal, hex-(2Z)-enal, hex-(2E)-enal, and pent-(2E)-enal could serve as indicators of unripe fruit, and many of the esters present could serve as indicators of ripeness, as in most cases, they were not detected in the unripe fruit. Based on the PLS-DA, discriminating ripening stages based on the aroma profile is challenging across cultivars, and it would be more accurate to create discrimination models separately for each cultivar. A similar conclusion can be drawn for the overall changes in the colour and secondary metabolite content. While ripening stages share similarities, unique indicators for each cultivar are more reliable due to qualitative differences in volatile organic and phenolic compounds. While ‘Malwina’ is suitable for processing due to its anthocyanin content, ‘Aprica’ and ‘Clery,’ with high ester content and a* values, are better suited for fresh consumption, emphasising aroma and colour.

## Figures and Tables

**Figure 1 plants-13-01419-f001:**
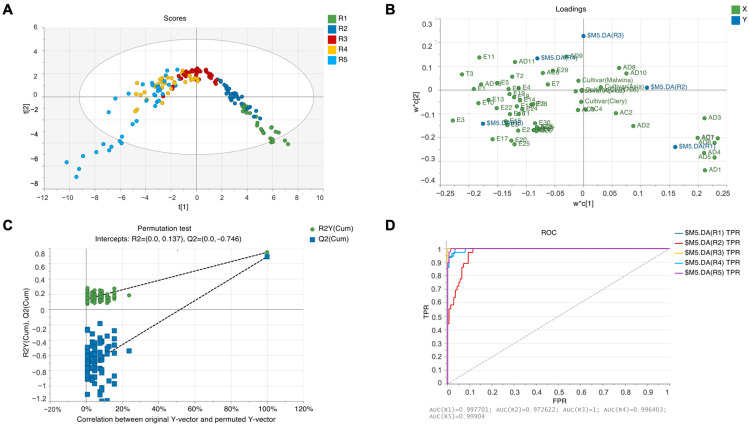
PLS-DA (**A**) score plot based on the first two components, (**B**) loading plot based on the first two components, (**C**) permutation test, and (**D**) ROC plot.

**Figure 2 plants-13-01419-f002:**
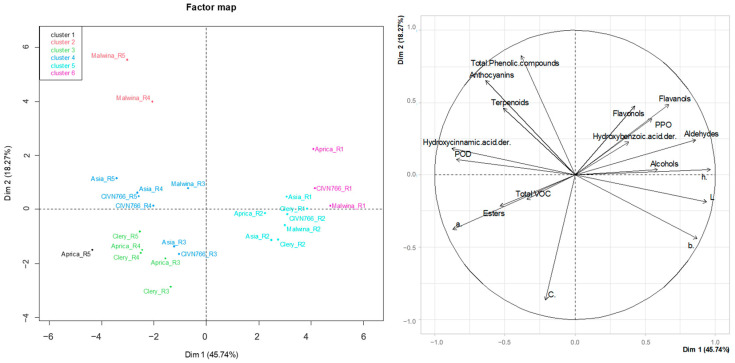
Principal component analysis map based on the colour parameters, contents of phenolic compounds, contents of volatile organic compounds, and peroxidase and polyphenol oxidase enzyme activity.

**Figure 3 plants-13-01419-f003:**
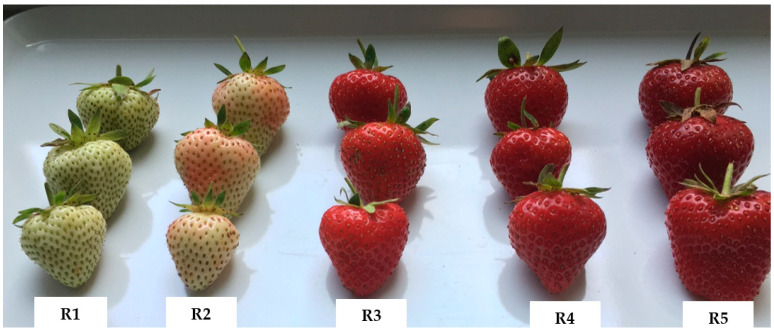
Five ripening stages of the fruit of the cultivar ‘Aprica’ (green fruit, R1; white fruit, R2; ripe red fruit, R3; fully ripe dark red fruit, R4; and overripe fruit, R5).

**Table 1 plants-13-01419-t001:** Colour parameters of the strawberry fruit at five different ripening stages of the studied strawberry cultivars.

Cultivar	Ripening Stage	Colour Parameters				
		L*	a*	b*	C*	h°
‘Aprica’	Green (R1)	53.7 ± 0.6 a	1.9 ± 0.3 d	30.9 ± 0.4 a	30.9 ± 0.4 c	86.5 ± 0.6 a
	White (R2)	56.5 ± 0.9 a	9.7 ± 1.6 c	27.4 ± 0.7 b	29.7 ± 0.5 c	70.5 ± 3.1 b
	Ripe (R3)	34.9 ± 0.7 b	38.6 ± 0.9 a	26.6 ± 0.9 b	46.9 ± 1.2 a	34.5 ± 0.6 c
	Fully ripe (R4)	32.6 ± 0.7 b	36.9 ± 1.1 ab	23.6 ± 0.9 c	44.4 ± 1.1 a	32.0 ± 0.7 cd
	Overripe (R5)	28.3 ± 2.0 c	34.3 ± 0.9 b	19.4 ± 0.8 d	39.5 ± 1.2 b	29.3 ± 0.4 d
‘Asia’	Green (R1)	56.1 ± 0.6 b	1.4 ± 0.3 c	31.3 ± 0.6 a	31.4 ± 0.6 bc	88.0 ± 0.9 a
	White (R2)	62.2 ± 0.7 a	3.4 ± 0.3 c	28.4 ± 0.3 b	28.6 ± 0.3 c	84.5 ± 1.1 b
	Ripe (R3)	33.2 ± 1.0 c	33.2 ± 0.8 a	24.4 ± 1.2 c	40.7 ± 1.4 a	36.0 ± 0.9 c
	Fully ripe (R4)	29.2 ± 0.7 d	28.6 ± 1.2 b	17.7 ± 1.0 d	33.7 ± 1.5 b	31.5 ± 0.7 d
	Overripe (R5)	27.3 ± 0.7 d	26.8 ± 1.4 b	15.3 ± 0.8 d	30.3 ± 1.3 c	30.3 ± 0.6 d
‘CIVN766’	Green (R1)	54.4 ± 0.8 b	2.0 ± 0.4 c	33.9 ± 0.6 a	34.0 ± 0.6 bc	86.7 ± 0.6 a
	White (R2)	60.7 ± 0.5 a	2.2 ± 0.3 c	32.2 ± 0.3 a	32.3 ± 0.3 c	86.1 ± 0.6 a
	Ripe (R3)	33.4 ± 0.9 c	35.6 ± 0.9 a	24.2 ± 0.9 b	43.5 ± 1.1 a	35.8 ± 2.1 b
	Fully ripe (R4)	29.2 ± 0.8 d	30.8 ± 1.2 b	18.4 ± 0.8 c	35.9 ± 1.4 b	30.8 ± 0.6 c
	Overripe (R5)	29.5 ± 0.3 d	30.7 ± 0.8 b	17.2 ± 0.6 c	35.1 ± 1.0 bc	29.2 ± 0.4 c
‘Clery’	Green (R1)	59.8 ± 0.5 a	1.3 ± 0.2 c	32.8 ± 0.5 a	32.8 ± 0.5 c	87.8 ± 0.4 a
	White (R2)	57.7 ± 1.6 a	13.6 ± 3.4 b	30.1 ± 0.5 b	34.8 ± 1.5 c	68.1 ± 5.2 b
	Ripe (R3)	35.2 ± 0.6 b	38.3 ± 1.1 a	28.9 ± 0.8 b	48.9 ± 1.0 a	36.2 ± 0.5 c
	Fully ripe (R4)	30.2 ± 0.6 c	32.7 ± 2.5 a	21.1 ± 0.6 c	41.3 ± 1.1 b	30.7 ± 0.4 c
	Overripe (R5)	30.2 ± 0.6 c	33.5 ± 1.1 a	19.1 ± 0.9 d	38.6 ± 1.4 b	29.6 ± 0.5 c
‘Malwina’	Green (R1)	54.7 ± 1.0 a	1.9 ± 0.3 d	35.5 ± 0.8 a	35.5 ± 0.8 a	86.9 ± 0.5 a
	White (R2)	52.5 ± 2.3 a	12.4 ± 3.3 c	30.6 ± 0.9 b	34.9 ± 1.2 a	69.5 ± 5.1 b
	Ripe (R3)	29.9 ± 1.6 b	28.0 ± 2.0 a	20.4 ± 2.2 c	34.8 ± 2.8 a	33.6 ± 1.5 c
	Fully ripe (R4)	22.3 ± 0.6 c	20.7 ± 0.8 b	11.9 ± 0.6 d	23.9 ± 1.0 b	29.8 ± 0.7 c
	Overripe (R5)	22.0 ± 0.5 c	15.2 ± 0.8 c	10.0 ± 0.5 d	17.6 ± 0.7 c	33.6 ± 1.8 c
Variance	Cultivar	31.9 ***	33.4 ***	25.1 ***	62.2 ***	5.2 ***
	Ripening stage	1144.3 ***	541.4 ***	350.9 ***	79.5 ***	1158.3 ***
	Cultivar: Ripening stage	4.9 ***	9.8 ***	11.5 ***	19.0 ***	5.2 ***

Different letters indicate statistically significant differences among the ripening stages of each cultivar separately (ANOVA, Duncan’s test, *p* < 0.05). ***: significant effect at *p* < 0.001.

**Table 2 plants-13-01419-t002:** Content of phenolic compounds (mg kg^−1^ dry weight) in the strawberry fruit at five different ripening stages.

Cultivar	Ripening Stage	Phenolic Compound Content (mg kg^−1^ Dry Weight)					
		Anthocyanins	Hydroxycinnamic Acid Der.	Flavanols	Hydroxybenzoic Acid Der.	Flavonols	Total
‘Aprica’	Green (R1)	nd	147 ± 9 d	6961 ± 305 a	5949 ± 293 a	461 ± 21 a	13,519 ± 606 a
	White (R2)	5 ± 1 c	269 ± 18 c	4219 ± 289 b	2024 ± 129 b	429 ± 25 a	6946 ± 449 b
	Ripe (R3)	3372 ± 234 b	777 ± 58 b	1346 ± 64 c	672 ± 46 c	466 ± 38 a	6631 ± 231 b
	Fully ripe (R4)	4462 ± 290 a	797 ± 33 b	1343 ± 52 c	608 ± 35 c	400 ± 43 a	7610 ± 222 b
	Overripe (R5)	4053 ± 457 ab	1233 ± 49 a	1522 ± 73 c	951 ± 60 c	290 ± 20 b	8048 ± 615 b
‘Asia’	Green (R1)	nd	79 ± 1 e	2627 ± 90 a	1673 ± 65 a	585 ± 48 a	4965 ± 143 d
	White (R2)	13 ± 1 c	266 ± 17 d	1754 ± 56 b	786 ± 24 c	226 ± 11 c	3044 ± 66 e
	Ripe (R3)	3307 ± 240 b	938 ± 27 c	1035 ± 47 d	456 ± 25 d	253 ± 6 c	5988 ± 293 c
	Fully ripe (R4)	5492 ± 319 a	1330 ± 75 b	1142 ± 34 cd	699 ± 33 c	284 ± 5 bc	8945 ± 288 b
	Overripe (R5)	5220 ± 286 a	1976 ± 101 a	1279 ± 53 c	1024 ± 22 b	338 ± 14 b	9836 ± 400 a
‘CIVN766’	Green (R1)	28 ± 4 c	114 ± 4 c	3389 ± 192 a	5267 ± 203 a	685 ± 64 a	9485 ± 275 b
	White (R2)	32 ± 4 c	193 ± 11 c	3094 ± 101 a	2370 ± 102 c	510 ± 19 b	6199 ± 182 c
	Ripe (R3)	2129 ± 267 b	994 ± 42 b	1172 ± 102 c	1913 ± 111 c	406 ± 25 bc	6614 ± 502 c
	Fully ripe (R4)	3314 ± 266 a	1975 ± 107 a	1462 ± 174 bc	3695 ± 273 b	403 ± 45 bc	10,849 ± 727 b
	Overripe (R5)	3496 ± 283 a	2169 ± 91 a	1643 ± 131 b	5085 ± 345 a	328 ± 18 c	12,721 ± 637 a
‘Clery’	Green (R1)	nd	110 ± 4 e	1729 ± 45 b	1745 ± 73 a	395 ± 30 a	3978 ± 127 d
	White (R2)	14 ± 1 c	403 ± 25 d	2464 ± 81 a	688 ± 30 d	284 ± 15 b	3854 ± 137 d
	Ripe (R3)	3460 ± 133 b	1057 ± 38 c	138 ± 15 d	770 ± 12 cd	292 ± 15 b	5717 ± 106 c
	Fully ripe (R4)	4142 ± 408 b	1521 ± 60 b	265 ± 25 cd	924 ± 87 c	298 ± 15 b	7150 ± 474 b
	Overripe (R5)	5074 ± 292 a	1805 ± 75 a	290 ± 32 c	1138 ± 53 b	389 ± 17 a	8696 ± 304 a
‘Malwina’	Green (R1)	5 ± 1 d	208 ± 11 d	5108 ± 131 a	2701 ± 30 a	540 ± 36 ab	8563 ± 84 c
	White (R2)	79 ± 6 d	329 ± 24 d	2786 ± 89 bc	694 ± 63 b	598 ± 36 a	4481 ± 182 d
	Ripe (R3)	4716 ± 469 c	985 ± 80 c	1958 ± 152 d	700 ± 66 b	549 ± 30 ab	8909 ± 496 c
	Fully ripe (R4)	10,662 ± 449 b	1246 ± 61 b	2423 ± 115 cd	738 ± 25 b	597 ± 28 a	15,664 ± 446 b
	Overripe (R5)	13,190 ± 964 a	1877 ± 111 a	3223 ± 283 b	757 ± 27 b	498 ± 20 b	19,546 ± 1252 a
Variance	Cultivar	112.7 ***	42.0 ***	249.7 ***	401.8 ***	58.2 ***	123.7 ***
	Ripening stage	345.0 ***	706.7 ***	406.0 ***	289.0 ***	28.5 ***	164.8 ***
	Cultivar:Ripening stage	35.4 ***	13.7 ***	44.4 ***	55.7 ***	8.4 ***	31.5 ***

Der., derivatives; nd, not detected. Different letters indicate statistically significant differences among the ripening stages of each cultivar separately (ANOVA, Duncan’s test, *p* < 0.05). ***, significant effect at *p* < 0.001.

**Table 3 plants-13-01419-t003:** Content of individual anthocyanins (mg kg^−1^ dry weight) in the strawberry fruit at five different ripening stages.

Cultivar	Ripening Stage	Anthocyanin Content (mg kg^−1^ Dry Weight)									
		Cy-3-*O*-gal	Cy-3-*O*-glc	Plg-3-*O*-glc	Plg-3-*O*-rut	Plg-3-*O*-ara	Cy-3-(6″malonyl)glc	Plg Der.	Plg-3-(6″malonyl)glc	5-Pyranoplg-3-glc	Plg-3-*O*-acetylglc
‘Aprica’	Green (R1)	nd	nd	nd	nd	nd	nd	nd	nd	nd	nd
	White (R2)	nd	nd	5 ± 1 c	nd	nd	nd	nd	nd	nd	nd
	Ripe (R3)	119 ± 10 a	21 ± 3 a	2389 ± 186 b	110 ± 10 b	nd	15 ± 1 b	5 ± 1 b	713 ± 40 b	nd	nd
	Fully ripe (R4)	131 ± 12 a	29 ± 3 a	3175 ± 266 a	142 ± 17 ab	nd	19 ± 2 ab	9 ± 2 ab	955 ± 31 a	nd	nd
	Overripe (R5)	130 ± 15 a	22 ± 5 a	2724 ± 356 ab	192 ± 23 a	nd	25 ± 3 a	18 ± 5 a	942 ± 75 a	nd	nd
‘Asia’	Green (R1)	nd	nd	nd	nd	nd	nd	nd	nd	nd	nd
	White (R2)	nd	nd	13 ± 1 c	nd	nd	nd	nd	nd	nd	nd
	Ripe (R3)	48 ± 3 b	19 ± 2 b	3008 ± 226 b	155 ± 9 b	nd	nd	nd	4 ± 1 c	nd	73 ± 8 a
	Fully ripe (R4)	115 ± 9 a	33 ± 3 a	4973 ± 289 a	272 ± 31 a	nd	nd	nd	11 ± 1 b	nd	87 ± 7 a
	Overripe (R5)	120 ± 14 a	36 ± 2 a	4688 ± 259 a	266 ± 16 a	nd	nd	nd	17 ± 2 a	nd	92 ± 10 a
‘CIVN766’	Green (R1)	5 ± 1 b	nd	23 ± 3 c	nd	nd	nd	nd	nd	nd	nd
	White (R2)	10 ± 1 b	nd	19 ± 3 c	nd	nd	3 ± 0 b	nd	nd	nd	nd
	Ripe (R3)	77 ± 9 a	15 ± 2 a	1444 ± 220 b	nd	nd	16 ± 2 a	4 ± 1 b	572 ± 43 b	nd	nd
	Fully ripe (R4)	83 ± 10 a	15 ± 2 a	2320 ± 215 a	nd	nd	19 ± 3 a	10 ± 1 a	868 ± 49 a	nd	nd
	Overripe (R5)	70 ± 8 a	14 ± 1 a	2388 ± 220 a	nd	nd	18 ± 2 a	10 ± 2 a	997 ± 58 a	nd	nd
‘Clery’	Green (R1)	nd	nd	nd	nd	nd	nd	nd	nd	nd	nd
	White (R2)	7 ± 0 b	nd	7 ± 1 c	nd	nd	nd	nd	nd	nd	nd
	Ripe (R3)	109 ± 8 a	20 ± 1 b	2557 ± 96 b	114 ± 9 b	nd	7 ± 1 b	5 ± 1 b	648 ± 30 b	nd	nd
	Fully ripe (R4)	109 ± 7 a	26 ± 2 ab	3153 ± 338 ab	132 ± 15 ab	nd	9 ± 1 b	9 ± 1 b	704 ± 62 b	nd	nd
	Overripe (R5)	121 ± 18 a	32 ± 4 a	3698 ± 251 a	169 ± 14 a	nd	19 ± 2 a	21 ± 4 a	1013 ± 62 a	nd	nd
‘Malwina’	Green (R1)	4 ± 1 c	nd	2 ± 0 d	nd	nd	nd	nd	nd	nd	nd
	White (R2)	9 ± 1 c	nd	66 ± 6 d	nd	nd	4 ± 1 c	nd	nd	nd	nd
	Ripe (R3)	181 ± 20 b	28 ± 4 c	3966 ± 413 c	195 ± 16 c	10 ± 1 c	9 ± 1 b	3 ± 1 b	314 ± 23 c	6 ± 1 c	5 ± 1 b
	Fully ripe (R4)	279 ± 21 a	55 ± 3 b	9151 ± 389 b	393 ± 30 b	22 ± 1 b	16 ± 2 a	8 ± 1 b	701 ± 27 b	21 ± 1 b	15 ± 0 a
	Overripe (R5)	324 ± 33 a	68 ± 5 a	11,328 ± 880 a	553 ± 41 a	36 ± 4 a	16 ± 2 a	20 ± 3 a	798 ± 24 a	35 ± 2 a	14 ± 0 a
Variance	Cultivar	62.2 ***	54.5 ***	131.0 ***	81.1 ***	na	11.9 ***	ns	213.3 ***	na	209.5 ***
	Ripening stage	111.2 ***	27.9 ***	294.5 ***	50.2 ***	na	34.8 ***	32.2 ***	67.5 ***	na	ns
	Cultivar:Ripening stage	16.3 ***	7.5 ***	39.2 ***	11.2 ***	na	2.9 **	ns	7.4 ***	na	ns

Gal, galactoside; glc, glucoside; rut, rutinoside; ara, arabinoside; plg, pelargonidin; cy, cyanidin; der., derivative; nd, not detected; na, not applicable; ns, not significant. Different letters indicate statistically significant differences among the ripening stages of each cultivar separately (ANOVA, Duncan’s test, *p* < 0.05). *** and **, significant effect at *p* < 0.001 and *p* < 0.01, respectively.

**Table 4 plants-13-01419-t004:** Enzyme activity of peroxidase (POD) and polyphenol oxidase (PPO) (a min^−1^ mg^−1^ protein) in the strawberry fruit at five different ripening stages.

Cultivar	Ripening Stage	Enzyme Activity (A min^−1^ mg^−1^ Protein)	
		POD	PPO
‘Aprica’	Green (R1)	4.0 ± 0.2 c	0.62 ± 0.05 a
	White (R2)	3.9 ± 0.1 c	0.44 ± 0.02 b
	Ripe (R3)	5.5 ± 0.2 b	0.45 ± 0.02 b
	Fully ripe (R4)	5.4 ± 0.1 b	0.41 ± 0.03 b
	Overripe (R5)	6.9 ± 0.4 a	0.32 ± 0.02 c
‘Asia’	Green (R1)	3.1 ± 0.2 c	0.72 ± 0.03 a
	White (R2)	3.8 ± 0.3 b	0.67 ± 0.11 a
	Ripe (R3)	4.5 ± 0.3 a	0.57 ± 0.03 ab
	Fully ripe (R4)	5.0 ± 0.2 a	0.42 ± 0.02 bc
	Overripe (R5)	5.2 ± 0.2 a	0.34 ± 0.02 c
‘CIVN766’	Green (R1)	2.6 ± 0.3 d	0.47 ± 0.02 a
	White (R2)	3.3 ± 0.2 c	0.48 ± 0.02 a
	Ripe (R3)	4.6 ± 0.2 b	0.37 ± 0.02 b
	Fully ripe (R4)	4.9 ± 0.2 b	0.30 ± 0.02 b
	Overripe (R5)	6.2 ± 0.1 a	0.34 ± 0.03 b
‘Clery’	Green (R1)	2.2 ± 0.2 d	0.96 ± 0.05 a
	White (R2)	2.6 ± 0.1 d	0.68 ± 0.04 b
	Ripe (R3)	3.1 ± 0.2 c	0.36 ± 0.03 c
	Fully ripe (R4)	4.1 ± 0.1 b	0.33 ± 0.02 c
	Overripe (R5)	4.9 ± 0.2 a	0.27 ± 0.01 c
‘Malwina’	Green (R1)	2.8 ± 0.2 c	0.47 ± 0.04 b
	White (R2)	3.0 ± 0.2 c	0.63 ± 0.05 ab
	Ripe (R3)	4.0 ± 0.3 b	0.62 ± 0.07 ab
	Fully ripe (R4)	5.0 ± 0.2 a	0.66 ± 0.04 a
	Overripe (R5)	5.3 ± 0.3 a	0.72 ± 0.08 a
Variance	Cultivar	38.5 ***	26.7 ***
	Ripening stage	134.2 ***	40.2 ***
	Cultivar: Ripening stage	2.2 *	12.8 ***

Different letters indicate statistically significant differences among the ripening stages of each cultivar separately (ANOVA, Duncan’s test, *p* < 0.05). *** and *, significant effect at *p* < 0.001 and at *p* < 0.05, respectively.

**Table 5 plants-13-01419-t005:** Contents of volatile organic compounds (µg g^−1^ dry weight) in the strawberry fruit at five different ripening stages.

Cultivar	Ripening Stage	Content of Volatile Organic Compounds (µg g^−1^ Dry Weight)					
		Esters	Aldehydes	Alcohols	Terpenoids	Furanones	Total
‘Aprica’	Green (R1)	nd	64.0 ± 3.3 a	0.85 ± 0.03 a	0.11 ± 0.01 b	nd	65.0 ± 3.3 c
	White (R2)	nd	36.8 ± 2.6 b	0.43 ± 0.03 b	0.14 ± 0.01 b	nd	37.4 ± 2.6 d
	Ripe (R3)	35.0 ± 1.1 c	22.8 ± 1.4 c	0.32 ± 0.04 b	0.20 ± 0.02 b	nd	58.3 ± 2.3 cd
	Fully ripe (R4)	80.9 ± 5.6 b	25.9 ± 1.0 c	0.43 ± 0.03 b	0.39 ± 0.04 a	nd	107.5 ± 5.8 b
	Overripe (R5)	143.1 ± 14.1 a	15.3 ± 1.7 d	0.36 ± 0.05 b	0.47 ± 0.05 a	nd	159.2 ± 15.7 a
‘Asia’	Green (R1)	nd	43.0 ± 2.7 a	0.42 ± 0.04 a	nd	nd	43.4 ± 2.7 b
	White (R2)	0.2 ± 0.0 d	33.6 ± 2.5 b	0.38 ± 0.03 a	0.12 ± 0.02 b	nd	34.3 ± 2.5 bc
	Ripe (R3)	3.5 ± 0.3 c	23.2 ± 2.7 c	0.33 ± 0.03 a	0.27 ± 0.04 b	nd	27.3 ± 3.0 c
	Fully ripe (R4)	16.5 ± 1.3 b	24.9 ± 2.6 c	0.39 ± 0.06 a	0.78 ± 0.09 a	nd	42.6 ± 3.7 b
	Overripe (R5)	35.2 ± 2.1 a	21.2 ± 3.2 c	0.37 ± 0.03 a	0.86 ± 0.07 a	nd	57.6 ± 4.9 a
‘CIVN766’	Green (R1)	0.1 ± 0.0 c	42.7 ± 1.9 a	0.94 ± 0.07 a	nd	nd	43.8 ± 2.0 a
	White (R2)	0.5 ± 0.1 c	34.8 ± 2.1 b	0.49 ± 0.04 b	nd	nd	35.8 ± 2.2 b
	Ripe (R3)	5.3 ± 0.4 b	21.6 ± 1.5 c	0.39 ± 0.05 b	0.11 ± 0.01 b	nd	27.3 ± 1.5 c
	Fully ripe (R4)	14.3 ± 0.5 a	20.3 ± 1.1 c	0.47 ± 0.05 b	0.20 ± 0.02 a	0.149 ± 0.020 a	35.4 ± 1.6 b
	Overripe (R5)	15.8 ± 0.8 a	14.6 ± 0.8 d	0.43 ± 0.04 b	0.15 ± 0.01 b	0.154 ± 0.017 a	31.1 ± 1.5 bc
‘Clery’	Green (R1)	0.2 ± 0.0 b	43.4 ± 1.0 a	0.79 ± 0.04 ab	nd	nd	44.4 ± 1.0 b
	White (R2)	0.6 ± 0.0 b	40.6 ± 2.6 a	0.88 ± 0.14 a	nd	nd	42.1 ± 2.7 b
	Ripe (R3)	42.1 ± 4.0 a	22.7 ± 0.9 b	0.55 ± 0.06 bc	0.24 ± 0.02 a	nd	65.6 ± 4.7 a
	Fully ripe (R4)	51.8 ± 5.1 a	20.2 ± 1.6 b	0.45 ± 0.04 c	0.27 ± 0.02 a	nd	72.7 ± 4.9 a
	Overripe (R5)	53.5 ± 4.7 a	19.4 ± 1.9 b	0.96 ± 0.13 a	0.29 ± 0.03 a	nd	74.2 ± 4.5 a
‘Malwina’	Green (R1)	nd	73.2 ± 3.0 a	1.12 ± 0.02 a	0.09 ± 0.01 d	nd	74.4 ± 3.0 a
	White (R2)	0.1 ± 0.0 d	47.7 ± 3.5 b	0.58 ± 0.06 b	0.10 ± 0.02 d	nd	48.5 ± 3.6 c
	Ripe (R3)	7.1 ± 1.6 c	31.9 ± 2.2 cd	0.45 ± 0.02 c	0.35 ± 0.03 c	nd	39.8 ± 1.1 d
	Fully ripe (R4)	14.7 ± 1.0 b	33.9 ± 1.9 c	0.46 ± 0.02 c	0.52 ± 0.05 b	nd	49.6 ± 2.8 c
	Overripe (R5)	29.6 ± 0.9 a	26.6 ± 1.3 d	0.38 ± 0.02 c	0.71 ± 0.06 a	nd	57.3 ± 2.2 b
Variance	Cultivar	135.7 ***	40.5 ***	29.4 ***	69.4 ***	na	99.9 ***
	Ripening stage	167.7 ***	194.5 ***	40.3 ***	115.7 ***	na	46.2 ***
	Cultivar: Ripening stage	32.5 ***	5.7 ***	7.6 ***	10.6 ***	na	20.3 ***

Nd, not detected; na, not applicable. Different letters indicate statistically significant differences among the ripening stages of each cultivar separately (ANOVA, Duncan’s test, *p* < 0.05). ***: significant effect at *p* < 0.001.

## Data Availability

All data are presented in the manuscript and the Appendix A.

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
