# Peer review of "Changes in the Aroma Profile and Phenolic Compound Contents of Different Strawberry Cultivars during Ripening"

_plants, 2024, doi:10.3390/plants13101419_

Round 1
Reviewer 1 Report
Comments and Suggestions for Authors
The manuscript entitled 'How does the profile of aroma and phenolic compounds in strawberries change during ripening?' needs a minor revision.
The main notes:
1. The title of the manuscript could be rephrased - for instance, into: 'Changes in the profile of aroma and phenolic compounds in different strawberry cultivars during ripening'.
2.A minor check is required for the English language and style through the text. Thus, in the Abstract, the authors enclose parts of sentences in parentheses three times. In my opinion, they should be rephrased a little.
3. The word 'cultivar' should be added to the list of keywords, because it occurs dozens of times in the text.
4. It needs to improve the objectives and rationale of the study in the end of the Introduction section. I believe that the purpose of the research (lines 57-66) should be formulated more succinctly.
5. Line 341: the name of method HPLC should be deciphered. It is also worth describing the methodology of the analysis in more detail.
6. In the "Methods" section, it is necessary to indicate the origin of the investigated cultivars (who and when they were created) and their localization during cultivation, since soil and climatic conditions have a significant effect on the accumulation of secondary metabolites
Comments on the Quality of English LanguageA minor check is required for the English language and style through the text.
Author Response
Dear Reviewer,
Thank you for your valuable comments.
Please find answers to you individual comments below.
- The title of the manuscript could be rephrased - for instance, into: 'Changes in theprofile of aroma and phenolic compounds in different strawberry cultivarsduring ripening'.
The title of the manuscript was updated.
- A minor check is required for the English language and style through the text. Thus, in the Abstract, the authors enclose parts of sentences in parentheses three times. In my opinion, they should be rephrased a little.
The abstract was updated to remove some of the parentheses. The manuscript was also sent for English editing.
- The word 'cultivar' should be added to the list of keywords, because it occurs dozens of times in the text.
The word was added to the list of keywords.
- It needs toimprove the objectives and rationale of the studyin the end of the Introduction section. I believe that the purpose of the research (lines 57-66) should be formulated more succinctly.
The last paragraphs including the objectives of this study was updated to clarify the objective of this study.
- Line 341: the name of method HPLCshould be deciphered. It is also worth describing the methodology of the analysis in more detail.
Additional details regarding the HPLC analysis was added.
- In the "Methods" section, it is necessary to indicate the origin of the investigated cultivars (who and when they were created) and their localization during cultivation, since soil and climatic conditions have a significant effect on the accumulation of secondary metabolites
More details regarding the origin of the cultivars and growing conditions were added to the 4.1. Plant Material section.
Reviewer 2 Report
Comments and Suggestions for Authors
This study focuses on the changes in the content of phenolic compounds and volatile organic compounds during ripening. The general quality of the paper is good, but there are still some areas that need revising or consideration.
1. In line 10: Consistently use "volatile organic compounds" throughout the manuscript, including in the keywords, to eliminate ambiguity and align with standard terminology.
2. In line 36, “however” means a shift in the trend of events, It doesn't make logical sense to put it here.
3. In line 51, the separation appears unnecessary. Integrating this content with the preceding paragraph could improve the flow and coherence of the introduction.
4. In line 52, some references (13-18) are outdated; suggest citing “Plant volatile organic compound (E)-2-hexenal facilitates Botrytis cinerea infection of fruits by inducing sulfate assimilation” and “Comparative analysis of volatile compounds in different muskmelon cultivars in Xinjiang based on HS-SPME-GC-MS and transcriptomics”.
5. In line 68, color analysis is limited to trends; more information can be obtained.
6. Line 127: Why analyze "anthocyanin content" alone?
7. There was relatively good discussion on the accumulation and metabolism of polyphenols, but less on VOCs.
8. Although the paper presents an extensive dataset underscoring the distinct nutritional and sensory qualities of different strawberry varieties, it falls short of delivering conclusive insights. A more decisive synthesis of the findings is recommended to underscore their implications.
Comments on the Quality of English LanguageNo comment.
Author Response
Dear Reviewer,
Thank you for your valuable comments. Please find answers to your individual comments below.
- In line 10: Consistently use "volatile organic compounds" throughout the manuscript, including in the keywords, to eliminate ambiguity and align with standard terminology.
- The terminology was updated throughout the text and also in the list of keywords.
- In line 36, “however” means a shift in the trend of events, It doesn't make logical sense to put it here.
- The sentence was rephrased to make logical sense.
- In line 51, the separation appears unnecessary. Integrating this content with the preceding paragraph could improve the flow and coherence of the introduction.
- The sentence was rephrased and this paragraph was connected with the previous paragraph.
- In line 52, some references (13-18) are outdated; suggest citing “Plant volatile organic compound (E)-2-hexenal facilitates Botrytis cinerea infection of fruits by inducing sulfate assimilation” and “Comparative analysis of volatile compounds in different muskmelon cultivars in Xinjiang based on HS-SPME-GC-MS and transcriptomics”.
- Additional references from the years 2021-2024 related to changes in aroma profile of strawberries during ripening were added.
- In line 68, color analysis is limited to trends; more information can be obtained.
- Additional observations regarding the absolute values of the colour parameters were added to the Results section.
- Line 127: Why analyze "anthocyanin content" alone?
The anthocyanin content was analysed as part of the phenolic content analysis. The title of the 4.5 section was updated to clarify this.
The content of individual anthocyanins was presented as they are the major phenolic compounds group present in ripe strawberries and contribute to the characteristic colour of the fruit.
- There was relatively good discussion on the accumulation and metabolism of polyphenols, but less on VOCs.
The discussion regarding the synthesis and accumulation of VOCs was extended.
- Although the paper presents an extensive dataset underscoring the distinct nutritional and sensory qualities of different strawberry varieties, it falls short of delivering conclusive insights. A more decisive synthesis of the findings is recommended to underscore their implications.
The conclusion section was updated to provide recommendations for the future.
Reviewer 3 Report
Comments and Suggestions for Authors
Comments to the authors:
The content of the work is interesting. Nevertheless, the work needs revision and some issues should be considered before the work could be considered for publication. The followings are some comments and suggestions for authors to consider and improve the manuscript.
1. I have found some grammar errors in the manuscript. The authors needed to carefully check throughout the manuscript and correct the grammar errors.
2. Novelty of the work should be clearly stated.
3. How did the RIs of volatiles calculated? Please provide the detail method. In addition, the RI of the compounds (including Retention index of compounds in reference) should be provided in the manuscript.
4. It would be more interesting if the authors could provide the key differential compounds to distinguish the flavor profiles of strawberry fruit at different ripening stages. Multivariate statistical analysis (e.g., partial least squares–discriminant analysis) is required for screening the different metabolites.
5. The odor description of volatiles should be provided in Table S4.
6. The data also need in-depth analysis and more discussion is needed for the results obtained. The current forms are not acceptable on their own. Specially, discussion should be more meaningful with proper translations of the obtained data, based on scientific evidences. Also, please compare the findings with previous research results. For instance, the reason for the change of flavor profiles should be discussion in detail.
7. The latin names should be in italic. For instance, reference 17, Fragaria x Ananassa.
Comments on the Quality of English LanguageExtensive editing of English language required.
Author Response
Dear Reviewer,
Thank your valuable comments. Please find answers to your individual comments below.
- I have found some grammar errors in the manuscript. The authors needed to carefully check throughout the manuscript and correct the grammar errors.
The manuscript was sent for English editing and changes are reflected in the new revised file.
- Novelty of the work should be clearly stated.
An additional sentence noting the difference compared to other studies was added to the Introduction.
Most previous studies focus on one cultivar, a limited number of ripening stages or one group of secondary metabolites. Therefore, there is a need to study these overall changes in the content of secondary metabolites in more cultivars grown under the same conditions in order to understand these changes and identify markers which can help monitor the ripening process and determine an optimal harvest point
- How did the RIs of volatiles calculated? Please provide the detail method. In addition, the RI of the compounds (including Retention index of compounds in reference) should be provided in the manuscript.
It was an error, we were measuring only the retention times (Rt). The methodology part was updated.
The retention times were added to the tables in the Supplementary Material S4.
- It would be more interesting if the authors could provide the key differential compounds to distinguish the flavor profiles of strawberry fruit at different ripening stages. Multivariate statistical analysis (e.g., partial least squares–discriminant analysis) is required for screening the different metabolites.
PLS-DA analysis was performed, and results are presented in Figure 1 and described in the last two paragraphs of section 2.4.
- The odor description of volatiles should be provided in Table S4.
Sensory descriptors were provided based on available database, that are note in the methodology.
- The data also need in-depth analysis and more discussion is needed for the results obtained. The current forms are not acceptable on their own. Specially, discussion should be more meaningful with proper translations of the obtained data, based on scientific evidences. Also, please compare the findings with previous research results. For instance, the reason for the change of flavor profiles should be discussion in detail.
The discussion regarding VOCs was extended to provide more information on the synthesis and accumulation of the compounds based on the available literature.
- The latin names should be in italic. For instance, reference 17, Fragaria x Ananassa.
The Latin names are written in italics.
Reviewer 4 Report
Comments and Suggestions for Authors
A well presented research paper, with data clearly presented and easy to read. To add value the research team should consider adding the following aspects
1. Include the growing conditions rather than just referencing. Ensure that the temperature and lighting environment is included indicating any diurnal variations, so tha if the research is followed at the grower level all aspects are covered.
2. Make reference if initial fruiting has a chlorophyl stage and when this declines in association with other pigmemts
3. Although a technical paper emphasising secondary metabolites, include reference to physical changes that are also quality characters. ie for the consumer palatability and texture are quality characters
4. In the discussion indicate or make reference to how sugar and acid levels parallel pigment stages.
Author Response
Dear Reviewer,
Thank you for your valuable comments. Please find the answers to your individual comments below.
- Include the growing conditions rather than just referencing. Ensure that the temperature and lighting environment is included indicating any diurnal variations, so tha if the research is followed at the grower level all aspects are covered.
- More details have been added to the 4.1. Plant Material section about the growing conditions.
- Make reference if initial fruiting has a chlorophyl stage and when this declines in association with other pigmemts
- Reference to chlorophyll content of the fruit was added at the begging of the third paragraph of the Discussion.
- Although a technical paper emphasising secondary metabolites, include reference to physical changes that are also quality characters. ie for the consumer palatability and texture are quality characters
- Note was added at the end of the discussion.
- In the discussion indicate or make reference to how sugar and acid levels parallel pigment stages.
- Note was added at the end of the discussion.